# Cerebrospinal Fluid Metabolomics: Pilot Study of Using Metabolomics to Assess Diet and Metabolic Interventions in Alzheimer’s Disease and Mild Cognitive Impairment

**DOI:** 10.3390/metabo13040569

**Published:** 2023-04-17

**Authors:** Angela J. Hanson, William A. Banks, Lisa F. Bettcher, Robert Pepin, Daniel Raftery, Sandi L. Navarro, Suzanne Craft

**Affiliations:** 1Division of Gerontology and Geriatric Medicine, Department of Medicine, University of Washington, Seattle, WA 98104, USA; 2Geriatrics Research Education and Clinical Center, Veterans Affairs Puget Sound Health Care System, Seattle, WA 98102, USA; 3Department of Anesthesiology and Pain Medicine, Northwest Metabolomics Research Center, University of Washington, Seattle, WA 98109, USA; 4Division of Public Health Sciences, Fred Hutchinson Cancer Center, Seattle, WA 98109, USA; 5Department of Internal Medicine, Wake Forest School of Medicine, Winston-Salem, NC 27109, USA

**Keywords:** metabolomics, ketone bodies, cerebrospinal fluid, Alzheimer’s, blood brain barrier

## Abstract

Brain glucose hypometabolism is an early sign of Alzheimer’s disease (AD), and interventions which offset this deficit, such as ketogenic diets, show promise as AD therapeutics. Conversely, high-fat feeding may exacerbate AD risk. We analyzed the metabolomic profile of cerebrospinal fluid (CSF) in a pilot study of older adults who underwent saline and triglyceride (TG) infusions. Older adults (12 cognitively normal (CN), age 65.3 ± 8.1, and 9 with cognitive impairment (CI), age 70.9 ± 8.6) underwent a 5 h TG or saline infusion on different days using a random crossover design; CSF was collected at the end of infusion. Aqueous metabolites were measured using a targeted mass spectroscopy (MS) platform focusing on 215 metabolites from over 35 different metabolic pathways. Data were analyzed using MetaboAnalyst 4.0 and SAS. Of the 215 targeted metabolites, 99 were detectable in CSF. Only one metabolite significantly differed by treatment: the ketone body 3-hydroxybutyrate (HBA). Post hoc analyses showed that HBA levels were associated with age and markers of metabolic syndrome and demonstrated different correlation patterns for the two treatments. When analyzed by cognitive diagnosis group, TG-induced increases in HBA were over 3 times higher for those with cognitive impairment (change score CN +9.8 uM ± 8.3, CI +32.4 ± 7.4, *p* = 0.0191). Interestingly, individuals with cognitive impairment had higher HBA levels after TG infusion than those with normal cognition. These results suggest that interventions that increase plasma ketones may lead to higher brain ketones in groups at risk for AD and should be confirmed in larger intervention studies.

## 1. Introduction

Alzheimer’s disease (AD) is the most common cause of dementia. One AD risk factor identified in epidemiologic studies is a ‘Western diet’, which is characterized by high intake of saturated high-fat foods (HFF) [1,2]. Animal experiments support this association, as HFF increases amyloid breakdown products, induces brain inflammation, and reduces brain insulin levels, all of which are pathologic features of AD [3,4,5,6]. Remarkably, a single high-fat meal increased markers of peripheral inflammation and oxidation in human studies, [7,8] and worsened cognitive test results in older adults [9]. Triglyceride (TG) infusions have also been used to study and mimic dyslipidemic insulin resistant states [10], and a 4 h TG infusion in healthy adults adversely affected hippocampal energy metabolism [11]. We recently published the lipidomic signature of cerebrospinal fluid (CSF) in older adults after a TG infusion using a common lipid component of total parenteral nutrition, finding significant changes in lipid classes as well as individual lipids [12]. This study endeavored to characterize CSF metabolomic changes from the same experimental cohort and approach.

Another consequence of AD is brain glucose hypometabolism [13]. Despite the discovery of insulin function in the brain, brain glucose uptake is thought to be widely insulin independent [3]. When the brain is unable to utilize glucose, such as during diabetic ketoacidosis, neurons can take up and utilize ketone bodies derived from plasma free fatty acids. Ketones in a variety of forms are being studied as a potential AD therapeutic, including ketone diets which are often high in fat content [13]. Given the above concerns about high-fat diet as a possible pathway to AD, understanding lipid metabolism in older adults—particularly those at risk for cognitive impairment or metabolic syndrome—will be important as these treatments become more widely utilized.

Analysis of the CSF metabolome can offer biochemical insights into brain function [14]. This body fluid is in direct contact with the extracellular space of the brain and is thus amenable for metabolic profiling for various diseases and brain pathologies [15]. Several groups have characterized the lipidome and metabolome of human CSF and found over 300 detectable metabolites, with significant changes in the metabolite profile of AD patients compared with controls [14,16]. Unlike the lipidomic analysis which showed multiple differences in lipid species in CSF after the infusion [12], we identified only one metabolite which significantly differed in response to this intervention—the ketone body 3-hydroxybutyrate (HBA). Nevertheless, HBA levels show some interesting relationships with age, cognitive diagnosis, and features of metabolic syndrome. Given the importance of glucose metabolism to AD pathology, we also report on the metabolite readouts of glycolysis and the TCA cycle.

## 2. Materials and Methods

### 2.1. Subjects

The Institutional Review Boards of the University of Washington and the Veterans Affairs Puget Sound Health Care System approved the study. Subjects were recruited from the Memory Disorders Clinic of the VA Puget Sound Health Care System, and through advertisements in local newspapers. All potential subjects underwent a physical examination, clinical laboratory testing, and APOE genotyping using PCR amplification with primer sequences and methods as per Hixson et al. [17]. Inclusion criteria were age ≥ 50 and in good physical health, and the cohort was enriched for those with mild cognitive impairment or Alzheimer’s disease. All participants underwent full neuropsychological evaluation and met NINCDS/ADRDA criteria for MCI or probable AD, as determined by consensus diagnosis conference. A diagnosis of MCI was based on significant cognitive decline from previous levels of functioning on two or more neuropsychological tests. This battery of tests has been used in a variety of studies to identify persons at risk for AD with either amnestic or multidomain MCI. Exclusion criteria included serious systemic illnesses (e.g., severe cardiovascular, respiratory, hepatic, or renal disease), diabetes, or current or previous use of hypoglycemic agents or insulin due to their effects on lipid metabolism, anticoagulant medications, as lumbar punctures would be contraindicated in this population, cholesterol-lowering medications due to their effects on lipid metabolism, and anti-psychotic, anti-depressant, anti-convulsant, anxiolytic, or sedative medications due to their effects on cognitive testing. Moreover, abnormal lipid metabolism was excluded, defined as low-density lipoprotein (LDL) > 190 mg/dL, triglycerides > 340 mg/dL, total cholesterol > 260 mg/dL, or a known pathologic lipid disorder such as chylomicronemia. Subjects were asked to refrain from using any non-steroidal anti-inflammatory medications (ibuprofen or aspirin) for 5 days prior to each infusion visit.

### 2.2. Infusion Intervention

This was a randomized cross-over design involving an intravenous saline versus a triglyceride infusion in older adults, followed by a lumbar puncture. All study subjects were blinded to the intervention and arrived at the research unit after a 12 h fast. Bilateral 20-gauge IVs were placed, and blood was drawn at 0 h and 5 h. The participants received a 5 h infusion of normal saline (0.9% 15 cc/h) or 20% triglyceride emulsion at 45 cc/h (Liposyn II for the first few participants and then Intralipid for the remainder) on different days in random order. The infusion consisted of emulsified fat particles of approximately 0.4 micron in diameter, similar to naturally occurring chylomicrons, and contained 10% safflower oil and 10% soybean oil as well as egg yolk phosphatides [12]. A total of 3 h after the TG infusion was stopped, the L4-L5 space was accessed with a 24-gauge Sprott spinal needle (Sprott, Toronto, ON, Canada; Pajunk, Geisingen, Germany), and 30 mL of CSF was withdrawn into sterile syringes. Samples were divided into aliquots in prechilled polyethylene tubes, immediately frozen with dry ice, and stored at −70 °C until assays were performed.

### 2.3. Metabolomics Preparation and Statistical Analysis

Metabolites were extracted by first thawing the CSF on ice and adding 300 μL of 80% (*vol*/*vol*) methanol in water (including isotope-labeled standards) to a 50 μL aliquot of the sample. After vortexing, samples were kept on ice for 20 min and then spun at 4 °C for 15 min at high speed. Supernatant (150 μL per sample) was transferred to a new tube and dried down by using a speed vacuum at 30 °C. Dried samples were stored at −80 °C.

Targeted liquid chromatography-tandem mass spectrometry (LC-MS/MS) was performed according to methods developed at the University of Washington’s Northwest Metabolomics Research Center as per Skill et al. [18,19]. The platform separated and targeted 215 known and verified aqueous metabolites representing over 35 different metabolic pathways using multiple reaction monitoring (MRM, New York, NY, USA) methodology. The targeted platform consisted of a CTC autosampler, Shimadzu Nexera 20 LC pumps, and an AB-Sciex 6500+ Triple Quadrupole mass spectrometer. Sample injections of 5 μL for positive ionization mode and 10 μL for negative ionization mode were used. Both chromatographic separations were performed using HILIC on a Waters BEH Amide column (150 × 2.1 mm, 2.5 μm; Waters Corp, Milford, MA, USA) with a flow rate of 0.3 mL/min. The mobile phase was composed of solvents A (10 mM ammonium acetate in 94.9% H_2_O, 3.0% (*vol*/*vol*) acetonitrile, 2.0% methanol, and 0.2% acetic acid) and B (10 mM ammonium acetate in 92.9% (*vol*/*vol*) acetonitrile, 4.9% H_2_O, 2.0% methanol, and 0.2% acetic acid). After the initial 2 min isocratic elution of 90% (*vol*/*vol*) B, the percentage of solvent B decreased to 50% (*vol*/*vol*) at t = 5 min. The composition of solvent B was maintained at 50% (*vol*/*vol*) for 4 min (t = 9 min), and then the percentage of B gradually went back to 90% (*vol*/*vol*) to prepare for the next injection. The metabolite identities were confirmed by spiking a pooled serum sample used for method development with mixtures of standard compounds. The extracted multiple reaction monitoring peaks were integrated by using MultiQuant 3.0 software (AB Sciex, Toronto, ON, Canada). Of the 215 metabolites targeted, up to 99 metabolites were detected. We also targeted and detected 34 isotope-labeled metabolites that were used to both monitor sample preparation and run conditions as well as to determine absolute concentrations for the corresponding endogenous metabolites. For quality control (QC), a sample of CSF pooled from the study samples and a pooled human serum were analyzed every 10 samples and at the beginning and end of the runs. For the pooled serum sample, 116 metabolites and 34 labeled internal standards were detected with a median CV value of 3.9%. For the pooled aliquots of the CSF samples, we detected up to 90 metabolites and 32 labeled internal standards with a median CV value of 4.5%.

Data were analyzed using MetaboAnalyst 4.0 online software [20] for data processing, normalization, and analysis by treatment, and where indicated, further analyses were conducted by group (i.e., by cognitive diagnosis) using t tests and Chi square tests in Excel and repeated measures ANOVA using SAS University (2018 edition). We determined whether each species had been previously characterized in the Human Metabolome Database (Version 4.0), which includes 450 species previously characterized in the CSF [21].

## 3. Results

### 3.1. Demographics and Plasma Data

Table 1 shows the cohort demographics, including fasting lipid and glucose values. Nearly half of the participants were women, and just over 40% were cognitively impaired (3 with Alzheimer’s disease and 6 with mild cognitive impairment, as determined by consensus diagnosis according to NINCDS/ADRDA criteria). In this group, the men were slightly older than the women, but the ages were similar across the cognitive groups. Eight of the participants carried the AD risk gene APOE4 (E4 positive). When separated by cognitive diagnosis, those with any cognitive impairment had lower LDL; other baseline values did not significantly differ (Table 1).

### 3.2. Baseline Metabolomic Findings

Of the 215 targeted metabolites, 99 were detectable in the CSF, along with 28 of the utilized 32 isotope-labeled internal standards for absolute quantification. Targeted and detected metabolites marked with an * are shown in the Appendix A. Of these 99 metabolites, 81 metabolites were present in all 42 samples, an additional 6 metabolites were present in at least 50% of the samples, and 12 metabolites were detected in less than 50% of the samples and thus excluded from further analyses (and marked with a #). A total of 116 possible metabolites were not detectable in any CSF samples (Appendix A, listed at bottom). Figure 1 shows the breakdown of detectable metabolites by six major categories. Further details about the metabolic pathways of all metabolites are listed in the Appendix A.

Next, we looked specifically at glycolysis and TCA cycle intermediates. We classified 35 metabolites as being involved in glycolysis or the TCA cycle. Of these metabolites, 19 were present in CSF and most of these were TCA cycle intermediates or sugars such as glucose. We detected only two glycolysis cycle intermediates: glucose 1-phosphate (G1P) was not present in any participant’s CSF after the saline infusion but was detected in two samples after TG infusion. Glyceraldehyde-3-phosphate (D-GA3P) was detected in all samples. In contrast, several TCA cycle intermediates were detected in all CSF samples including citrate, aconitate, succinate, fumarate, and oxaloacetate (Appendix A).

### 3.3. Effect of Intervention on Metabolites

An average of 88 different metabolites were detected in each CSF sample, and this did not differ between saline and TG infusion (Saline 87.6 ± 1.4, TG 87.7 ± 1.7). A total of 12 metabolites were present in <50% of samples and thus were excluded from MetaboAnalyst (marked with a # in the Appendix A). Examination of these 12 metabolites indicated that the missing data were equally distributed across the TG and saline groups.

Next, we analyzed whether metabolite levels differed by treatment using the paired T test feature in MetaboAnalyst. Data were log transformed using the natural logarithm, metabolites with greater than 50% missing were excluded, and data less than 50% missing were imputed using a small value. To examine the overall data patterns, we analyzed the data using PCA and hierarchical clustering analysis (HCA), which are both shown in Figure 2. The PCA plot (Figure 2a) indicates little separation between the two treatments; therefore, other supervised methods were not conducted. The heat map (Figure 2b) largely confirms the PCA pattern with few cells highlighted as significant overall. Interestingly, the heat map data does split into two main groups, and with only one exception the participants’ saline and TG conditions clustered together. The first group division was not explained by sample date or APOE status, but participants in group 2 were more likely to be male (78% group 2 vs. 36% group 1) and more likely to be cognitively impaired (56% group 2 vs. 27% group 1). Group 2 contained all three AD patients.

Using Benjamini–Hochberg robust correction for multiple comparisons, only one significant metabolite was identified: an increase in the ketone body 3-hydroxybutyrate (HBA) after the TG infusion (Table 2).

To understand what participant factors might influence HBA levels, we performed correlation analysis with several metabolic syndrome risk factors as well as age. Age positively correlated with fasting HBA, whereas systolic blood pressure positively correlated with TG-induced HBA and with HBA change score (Table 3). Further exploration showed that older men had higher HBA concentrations than women. Participant BMI and fasting plasma lipid levels did not correlate with fasting or TG-induced HBA. In addition, we conducted repeated measures analysis of variance (ANOVA) to explore the influence of categorical variables of sex, E4 status, and cognitive diagnosis on HBA measures. Sex and E4 status were not significant in the model, but the interaction term condition*cognitive diagnosis was significant (condition*diagnosis F 6.54, *p* = 0.0198). Examination of the means showed that fasting HBA was similar by diagnosis, but TG-infusion induced increases in HBA were nearly 3 times higher for those with any cognitive impairment (age adjusted difference between change score *p* = 0.02, Table 4, Figure 3). When cognitive diagnosis was separated into MCI and AD groups, we noted a stepwise increase from CN to MCI to AD in the change score, going from 9.8 to 24.9 to 46.2 µM (condition*diagnosis *p* = 0.03); however, numbers of MCI and AD subjects were very small (Table 4, Figure 3).

Next, we used the Pattern Hunter feature of MetaboAnalyst to examine which metabolites correlated with CSF HBA levels (Appendix A). This feature allows the generation of either Pearson’s or Spearman’s correlation matrix for an individual metabolite. Given our crossover design, metabolite correlations were evaluated separately by treatment. Appendix A lists the top 10 Pearson’s correlations for both the saline and the TG samples. Some correlations appear on both lists including the amino-acid-related metabolites glucuronate and acetylglycine. For the saline infusion samples, HBA was positively correlated with acetoacetate (Pearson CC + 0.575, *p* = 0.006). For the TG samples, HBA did not significantly correlate with acetoacetate but correlated with pyruvate (Pearson CC + 0.573, *p* = 0.007).

## 4. Discussion

In this group of older adults, of the 215 possible metabolites measured in this targeted platform, 99 were detectable in CSF in at least some samples, and 81 were detectable in all samples. By and large, metabolite levels did not vary between the two interventions within individuals, indicating that CSF metabolites remain relatively stable over a short intervention. However, the HCA of the data, as shown in the heat map of Figure 2b, suggest that in a larger sample size, there may be some patterns noted by sex and cognitive diagnosis. The only metabolite that significantly changed in CSF with the TG infusion was the ketone body HBA. This metabolite increased in the CSF after TG infusion, and factors that correlated with higher levels of HBA levels included older age, markers of metabolic syndrome, and cognitive impairment.

We detected 99 metabolites in CSF overall, with an average of 88 per sample. These findings were expected because, due to the nature of CSF production, this fluid contains many fewer detectable proteins and metabolites than plasma. For example, the same methodology run on plasma would typically measure about 120–140 metabolites [22]. In other studies, the total number of metabolites reported in untargeted approaches can be over 300 (many of which are unidentified) [14]; however, using more targeted approaches, similar numbers of metabolites are noted in CSF to this study. For example, one study using hydrophilic interaction chromatography (HILIC) time-of-flight MS found 74 metabolites in CSF [23], and another study found 73 metabolites using LC/MS [24]. An exploratory study of transcranial magnetic stimulation in 5 patients with depression identified 72 metabolites in their targeted approach; and similar to our study, most CSF metabolites remained stable over the treatment condition. Four metabolites in that study differed using a *p* value cutoff value of 0.05, and only one met a more stringent cutoff of *p* < 0.01 [25]. A study of a ketamine infusion in healthy volunteers showed that of 630 potential metabolites, 82 metabolites were identified in CSF when excluding those with quantitative values in less than 67% of the total samples. In this study, circulating plasma levels of some metabolites did not always mirror CSF levels. For example, branched-chain amino acids increased in the plasma of healthy volunteers post-ketamine but decreased in the CSF, indicating the importance of assaying CSF for metabolic markers of interest in brain conditions [26].

In terms of the types of metabolites noted, many CSF metabolites belonged to nucleic or amino-acid-related metabolites or precursors. In addition, a total of 116 metabolites were never detected in any of the CSF samples, representing 54% of detectable metabolites in our targeted profile. Unlike the lipidomics analysis from this same intervention, which revealed numerous lipids and classes of lipids which changed in response to the infusion [12], here we observed very little change in metabolite levels or content, similar to at least one other study [25]. Glycolysis intermediates were largely absent in CSF, except for glyceraldehyde-3-phosphate (D-GA3P), which was present in all samples, and two TG infusion samples that had detectable glucose 1-phosphate. GA3P is an important metabolic intermediate in both glycolysis and gluconeogenesis as well as tryptophan biosynthesis [21]. This contrasted with TCA cycle intermediates, as many were present in all samples. These findings were similar to another group’s study of glucose metabolism in CSF in hepatic encephalopathy patients [24]. It is interesting that none of these TCA intermediates differed between the fasting and high TG state, as one would expect TCA metabolites to be more prevalent after high fat feeding.

Moreover, in our study, glucose, pyruvate, and lactate were widely present, but lactose, mannose and galactose sugars were not detectable, consistent with these sugars requiring further processing before brain metabolism [27]. Many phosphorylated nucleotides were not detectable in CSF. Further examination of these types of findings, potentially with labeled carbon in addition to CSF metabolomics, may be important in identifying the downstream consequences of glucose hypometabolism and whether all pathways are affected equally or whether there is a disproportionate effect on, for example, pentose phosphate pathway metabolites [28]. One group is looking at just this phenomenon: several anesthetic and sedative agents increased CSF metabolites in the pentose phosphate pathway, whereas others had opposite effects, in a study of patients with subarachnoid hemorrhage who are vulnerable to developing agitation and delirium [29]. Understanding the nuances of the glucose metabolites may help with designing diets or supplements for individuals at risk for AD, such as E4 carriers who have early brain glycose hypometabolism even prior to cognitive symptoms [30]. For example, after a glucose challenge healthy young adult E4 carriers showed a marked increase in lactate, and a pathway analysis of the plasma metabolome suggested an increase in aerobic glycolysis. Parallel work in E4-expressing astrocytes given labeled glucose also showed higher lactate synthesis and an increase in aerobic glycolysis rather than utilizing the TCA cycle [31]. To ascertain whether this was happening in the brains of human subjects since glycolysis intermediates are largely absent from CSF, other methods would need to be deployed such as those utilizing radiolabeled glucose.

### 4.1. The Ketone Body HBA

Only one metabolite met our statistical threshold as differing between the two conditions—the ketone body HBA. HBA is one of several ketone bodies synthesized in the liver from acetyl-CoA and can be used as an energy source by the brain when blood glucose is low [13,21]. This compound is also a partial degradation product of branched-chain amino acids (primarily valine), and is metabolized by 3-hydroxybutyrate dehydrogenase [21]. At least some of the anti-epileptic activity of the ketogenic diet has been ascribed to direct and indirect effects of HBA [32,33]. Butyrate and to a lesser extent HBA have emerged as important players in the mediation of communication between the microbiota and the brain [34]. Ketone bodies applied to the brain side of the BBB in an in-vitro model promoted amyloid clearance while maintaining BBB integrity [35]. Specifically, the ketone HBA readily crossed the BBB and alleviated neuroinflammation and memory impairment in various rodent models of neural injury [36,37]

Diets and supplements which induce ketosis are being investigated as treatments for AD [13]. Healthy older adults appear to have a similar capacity to produce ketones and oxidize HBA compared with younger adults [38], and in healthy men undergoing prolonged fasting, plasma HBA levels correlated with age [39]. Older adults with MCI in a 6-month randomized controlled trial of a ketogenic drink demonstrated increased plasma ketone levels, increased brain ketone levels as assessed by PET scan, and improved cognitive outcomes in multiple domains [40]. These acute studies suggest that older adults can mount a ketogenic response to ketogenic diets or supplements, and our study supports these findings as those with cognitive impairment had a higher increase in CSF HBA in response to a TG infusion than those with normal cognition. We further noted that age correlated with fasting HBA, and systolic blood pressure and fasting glucose correlated with TG-induced levels of CSF HBA. Given that age, hypertension, and glucose intolerance are risk factors for AD, these results suggest that individuals with these risk factors can increase their brain ketone levels in response to directed therapies.

We recently showed that after a 6-week modified ketogenic diet study in older adults, lower plasma ketones were noted for participants with MCI compared with those with normal cognition, and we speculated this could be due to either greater ketone uptake into target tissues or decreased ability to generate ketones [41]. This speculation was supported by the observation that cerebral uptake of a labeled ketone (11C-acetoacetate) assessed with PET in a subset of participants from the trial was increased after the ketogenic diet compared with a low-fat diet [41]. Given the findings in this and the above studies, [38,39,40] greater ketone uptake seems likely, and further metabolomic studies could assist clinical researchers with such questions of uptake vs. utilization when faced with changing levels of a metabolite. For example, using the Pattern Hunter feature of MetaboAnalyst, we show that for the saline samples, HBA correlated with acetoacetate which is consistent with a prolonged fasting state producing HBA and other ketones by the liver. For the TG state, HBA did not correlate with acetoacetate but instead correlated with pyruvate, which is likely due to increased generated pyruvate through the TCA cycle and more complete conversion of acetoacetate to HBA. No correlations were noted for either condition with valine or other branched-chain amino acids which can break down to form HBA. Additional pathway analysis may allow for hypothesis generation in samples and situations where definitive proof of metabolic pathway activation cannot be obtained, as with animal studies, or when radioactive tracers are not available or feasible.

### 4.2. APOE and Metabolites

Unlike the lipidomic signature [12], we did not find any significant APOE-related changes in the metabolites’ response to the TG intervention, including differences in HBA levels. It is likely this finding reflects the remarkable stability of the metabolites overall across the two interventions as well as the variability in the small sample size, making any APOE-related change difficult to detect. E4 carriers demonstrate early brain glucose hypometabolism even before other neuropathologic hallmarks such as brain amyloid are measurable [30]. Despite this deficit, studies are mixed with respect to whether E4 carriers respond to ketone supplements and diets which may ameliorate the glucose defect of AD [13]. Other metabolic rescue attempts are being studied for this group. One study found that supplementation with anserine and carnosine (a dipeptide formed from beta-alanine and histidine) showed a higher cognitive benefit for E4 carriers. This supplement was shown to reverse blood vessel abnormalities, and prevented pericyte damage and RAGE expression in AD mouse models [42]. A cohort study showed that E4 carriers had higher serum FFAs, acylcarnitines, and altered TCA cycle intermediates which fit with the hypothesis that this group showed deficiencies in cerebral glucose metabolism and potentially favored oxidation of FFA for energy sources [43]. Using metabolomics in clinical trials such as utilized here may help to elucidate pathways or predict whether various supplements or diets work better for various groups at risk for AD. Nevertheless, our data suggest these studies may require larger sample sizes to detect differences in CSF metabolites in the same individuals over time.

### 4.3. Limitations

The findings of this study are limited by a small sample size, which particularly limits our ability to look for subgroup effects by sex, diagnosis, or APOE genotype. Early in the study we had to change to a different manufacturer for the lipid infusion; however, the two infusions had the same components per the manufacturer. Findings would be strengthened by plasma metabolite analysis (not currently available in our data set), as well as metabolic analysis of the infusion. For future studies such as this one, we recommend analysis of the infusion product itself to identify all potential compounds.

## 5. Conclusions

In this small pilot study, we showed that unlike the lipidomic signature, the CSF metabolic signature remained remarkably stable after a triglyceride infusion in older adults at risk for AD. Most TCA metabolites were present in CSF, whereas glycolysis intermediates were mostly absent. The ketone body HBA was the only significant metabolite that changed in response to TG infusion, and those with cognitive impairment showed a more significant increase in CSF HBA levels. Metabolite studies such as the one presented here are feasible to explore metabolic hypotheses about ketone bodies in human brain, and more generally may be useful for understanding brain fuel metabolism.

## Figures and Tables

**Figure 1 metabolites-13-00569-f001:**
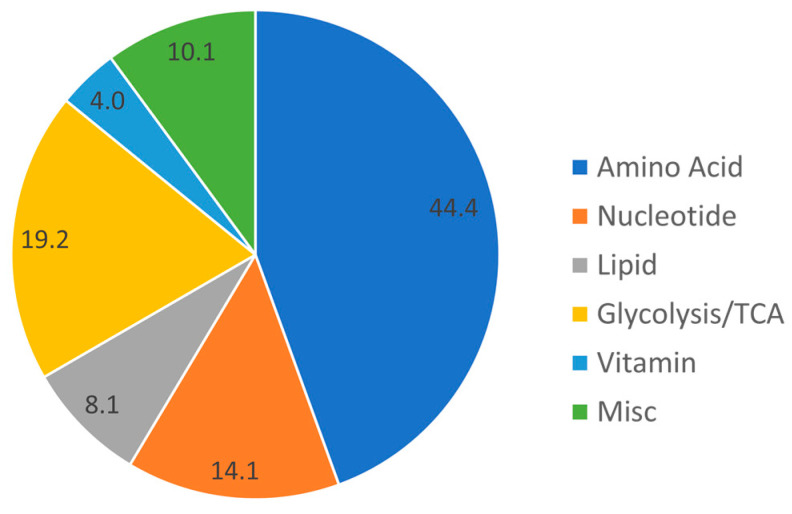
Pie chart of the metabolite categories present in the CSF.

**Figure 2 metabolites-13-00569-f002:**
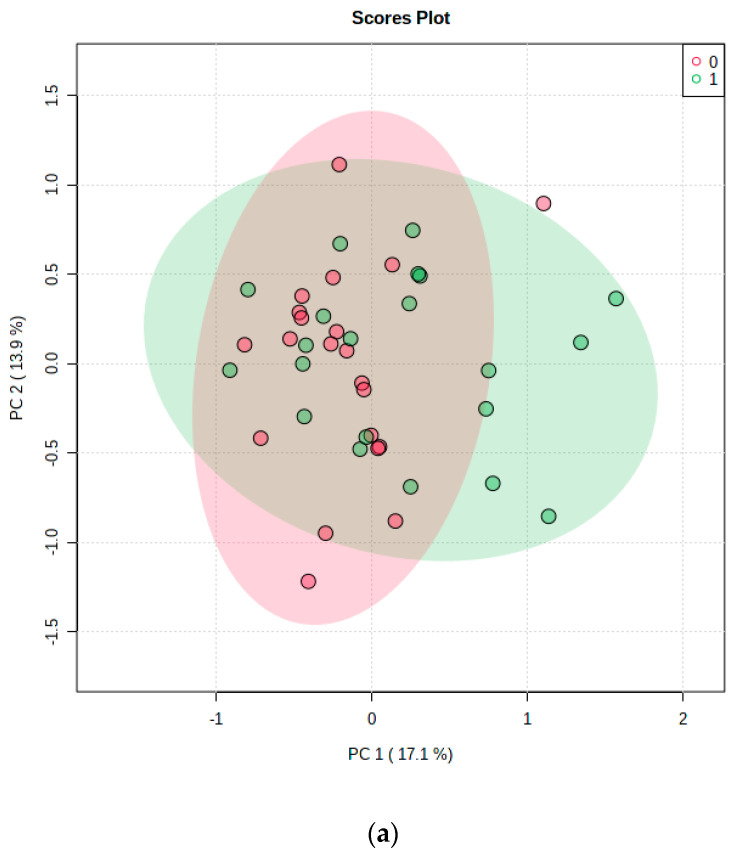
(**a**) PCA plot of the metabolites. Pink circles indicate saline; green circles indicate TG infusion. Graph was generated using MetaboAnalyst. (**b**) HCA heat map generated by Metaboanalyst (using Euclidean clustering, Ward method). Class 0 is saline; class 1 is TG infusion. Each colored cell on the map corresponds to a normalized concentration value, with samples in rows and compounds in columns. Red and blue colors indicate positive and negative correlations, respectively. Group 2 was more likely to be male and have more cognitive impairment.

**Figure 3 metabolites-13-00569-f003:**
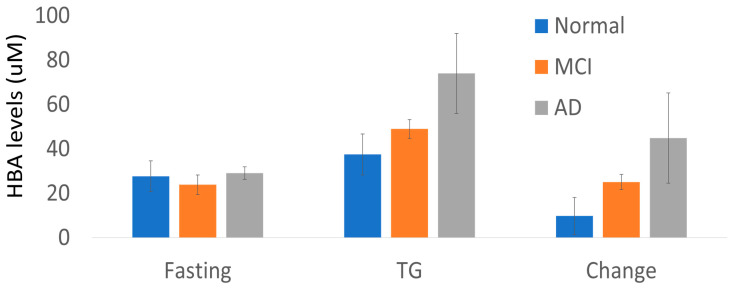
Quantitative HBA levels (uM) by cognitive diagnosis (adjusted for age) given are mean +/− standard error.

**Table 1 metabolites-13-00569-t001:** Demographics (mean plus standard deviation). # = number of subjects.

	All Subjects	CN	CI	T Test (Chi^2^)
Number	21	12	9	
# Female	10	7	3	*p* = 0.26
# E4 Pos	8	4	4	*p* = 0.6
Age	67.7 ± 8.6	65.3 ± 8.1	70.9 ± 8.6	*p* = 0.14
BMI (kg/m^2^)	25.7 ± 4.0	24.9 ± 3.6	26.8 ± 4.4	*p* = 0.29
HDL (mg/dL)	67.3 ± 19.1	70.9 ± 21.6	62.6 ± 14.8	*p* = 0.33
LDL (mg/dL)	116 ± 20.7	108.4 ± 18.8	126.1 ± 19.5	*p* = 0.049
Triglyceride	93.4 ± 33.2	97.8 ± 28.8	87.4 ± 39.5	*p* = 0.49
Glucose	90.4 ± 7.4	88.6 ± 6.7	92.8 ± 7.9	*p* = 0.20

CN = cognitively normal; CI = cognitive impairment.

**Table 2 metabolites-13-00569-t002:** Metabolite differences (conc in uM) between treatment groups. *p* value from paired T test.

Metabolite	Scheme	TG	*p* Value	Pathway
HBA	25.7 ± 17.6	45.6 ± 29.1	0.003	TCA Cycle

**Table 3 metabolites-13-00569-t003:** Pearson correlation matrix, cc (*p* value) with *p* < 0.05 bolded.

	Saline HBA	TG HBA	Change HBA (TG Minus Saline)
Age	**0.53 (0.013)**	0.35 (0.12)	0.028 (0.9)
Systolic BP	0.12 (0.6)	**0.5 (0.02)**	**0.46 (0.035)**
Fasting plasma glucose	0.24 (0.28)	**0.54 (0.012)**	0.42 (0.056)

**Table 4 metabolites-13-00569-t004:** HBA levels (uM) by cognitive diagnosis (adjusted for age) given are mean +/− standard error.

Cog Dx	Fasting	TG	Change
CN (n = 12)	27.7 ± 6.9	37.5 ± 9.2	9.8 ± 8.3
Cog (n = 9)	25.8 ± 3	58.2 ± 7.4	32.4 ± 7.4
MCI (n = 6)	23.9 ± 4.4	49 ± 4.2	25.1 ± 3.4
AD (n = 3)	29.1 ± 2.8	74 ± 18	44.9 ± 20.3

## Data Availability

The data that support the findings of this study are available from the corresponding authors upon reasonable request. Data is not publicly available due to privacy.

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
