# Peer review of "Cerebrospinal Fluid Metabolomics: Pilot Study of Using Metabolomics to Assess Diet and Metabolic Interventions in Alzheimer’s Disease and Mild Cognitive Impairment"

_metabolites, 2023, doi:10.3390/metabo13040569_

Round 1

Reviewer 1 Report

The paper by Hanson et al. examined the metabolite profile in the cerebrospinal fluid (CSF) in cognitively normal and cognitively impaired older adults following a 5-hour infusion of either saline or 20% triglyceride (TG) emulsion.  Of all the targeted metabolites examined, only hydroxybutyrate was altered by cognitive state, it being 3X higher in cognitively impaired.  The overarching research question that has conflicting reports is whether or not a ketogenic diet is neurological of benefit or harmful.  This paper does very little to advance the field.  How does the TG emulsion compare to a ketogenic diet or high-fat feeding?  What is the justification for the emulsion to be 10% safflower oil and 10% soybean oil?  Hydroxybutyrate is generated in the liver - an elevation in the CSF does not convey anything metabolic with respect to the brain.  Having 21 total subjects appears to be underpowered - was a power analysis done and if so what were the variables it was based on?

Author Response

Thank you for your questions and suggestions. We agree that 21 subjects is a small sample size which is why we titled this a feasibility study. We added extra language to the manuscript to emphasize that these findings are exploratory in nature. In terms of the emulsion composition: elevation of free fatty acids through TG emulsion is a well-studied endocrinologic model of the initial stages of insulin resistance (Boden et al 1994), and we and many researchers in this field use Liposyn (brand later changed to Intralipid) as this product is the standard lipid component of total parenteral nutrition (TPN). This product is safe and well tolerated in human subjects, contains a combination of saturated and unsaturated fatty acids, and is otherwise well characterized. Statements to this effect were added to the manuscript. Of note, this project did not set out to understand a ketogenic diet but rather was a part of a laboratory project to characterize the high lipid state in older adults as a known risk factor for Alzheimer's disease. When we found that the ketone body HBA increased in the CSF after a TG infusion—especially in those with cognitive impairment—we then chose to analyze this finding further given the current interest in ketones and Alzheimer’s. Since HBA has been shown in animal models and humans to be active in the CNS (we clarified this in the manuscript discussion section), we feel these findings are interesting regardless of the source of HBA. We do agree that a TG infusion is unlikely to be the therapeutic mechanism by which we would induce ketosis.

Reviewer 2 Report

Cognitive disorders are a global problem of modern medicine, which increases the importance of research aimed at studying the mechanisms of their development. The article is of research and clinical interest.

Comments:

1.      The title of the article suggests an analysis of metabolomics in Alzheimer's disease, but Alzheimer's disease has been diagnosed in only 3 patients. Is this number of patients sufficient? What was the cause of the cognitive impairment in the other patients?  

2.      Given the findings, it would be interesting to know the plasma levels of 3-hydroxybutyrate in these patients and their correlations with the levels in CSF. Especially since the abstract suggests that interventions that increase plasma ketone levels may lead to higher brain ketone levels in those at risk for AD. It would be logical to confirm these findings.

3.      It is recommended that all abbreviations be written in full when first mentioned. In materials and methods, it is recommended to add units of biochemical parameters in lines 374 and 375 and in Table 1.

Author Response

  1. Thanks for your question. We agree that the title is better clarified with the addition of Mild Cognitive Impairment and have made that change. In terms of whether these specific participants with MCI relate to Alzheimer’s disease: today we have sensitive methods including amyloid scans to better ensure enrollment of AD pathology; in the past we relied on clinical measures which were reasonably effective although not 100% perfect in identifying MCI that was likely AD. We have added this clarification in the methods section: “All participants underwent full neuropsychological evaluation and met NINCDS/ADRDA criteria for MCI or probable AD, as determined by consensus diagnosis conference. A diagnosis of MCI was based on significant cognitive decline from previous levels of functioning on two or more neuropsychological tests. This battery of tests has been used in a variety of studies to identify persons at risk for AD with either amnestic or multidomain MCI.”
  2. Agreed that this would be a great addition but unfortunately, we do not have access to the plasma samples and are unable to measure these in this cohort of subjects.
  3. Thank you, this has been done.

Reviewer 3 Report

This manuscript, “Cerebrospinal Fluid Metabolomics: Feasibility Study of Using Metabolomics to Assess Diet and Metabolic Interventions in Alzheimer’s Disease”, aimed to investigate how CSF metabolites change with TG infusion among elderly participants with or without cognitive impairment. Overall, the data analysis was appropriate, and the findings provided helpful insights.

The following comments or suggestions, if can be addressed, would further strengthen this manuscript.

  1. Figure 2: it would be helpful if the authors can add the height of the clustering tree. It will let readers know how well two groups were separated.
  2. Sample selection: Subjects using certain medications were excluded from the study population. Will it lead to potential selection bias?
  3. Authors discussed the potential confounding effects of age and metabolic states. Instead of this post-hoc approach, it could be helpful if authors can adjust these potential confounders within the linear model when conducing MWAS.
  4. LDL level is significantly different between CN and CI group. Did authors investigate the potential confounding effects of LDL?

Author Response

  1. This has been done and figure has been updated.

  1. Thank you for this question – always a concern with any inclusion and exclusion criteria. For this study, given the effects of medications on lipid transport into the brain, and on cognitive testing, we were concerned that certain classes of medications may blunt effects of the infusion or interfere with cognitive testing to assess for Alzheimer’s pathology. We better clarified why medications were excluded in the methods section: “Exclusion criteria included serious systemic illnesses (e.g., severe cardiovascular, respiratory, hepatic, or renal disease), diabetes or current or previous use of hypoglycemic agents or insulin due to their effects on lipid metabolism, anticoagulant medications as lumbar punctures would be contraindicated in this population, cholesterol-lowering medications due to their effects on lipid metabolism, and anti-psychotic, anti-depressant, anti-convulsant, anxiolytic, or sedative medications due to their effects on cognitive testing.”

  1. We have taken this advice and rerun some analyses using linear mixed models, with a statistician affiliated with our Metabolomics team. We ran additional models (linear mixed models) including age, sex and metabolic covariates, and essentially the results did not change. HBA was still highly statistically significant after the TG infusion with adjustment of our main covariates. I.e., we did find the same results when we included these in the model. Systolic BP was marginally significant at P=0.05 and glucose was significant at P=0.03. We also found that both age and sex were significantly associated with HBA, and additional exploration showed that older men had higher concentrations than older women. We have added a statement to this effect to the manuscript. Our statistician advised that proceeding with the reporting on correlation matrix was advisable given that the different groups are potentially giving us different information (here comparing fasting states to treatment states, etc) rather than running everything in the model together.

  1. We did include LDL and other plasma lipids as covariates to both our ANOVA and our linear mixed models, and it was not significant, nor did it change the point estimates.

Reviewer 4 Report

The manuscript metabolites-2199764 is interesting and within the journal's scope. I would like to see more detail in the experimental section because it is difficult to find the procedures used. The reference suggested by the authors directs us to other references. The authors could include detailed procedures in the supplementary information. Limitations and Conclusions should also be point 4 instead of 3.5. 

Author Response

We have corrected the number error.  In terms of the details of the methods, we apologize for difficulties in finding the procedures. We noted that the reference we used also gave another reference for the basic methods used. We have included the reference that has the most detailed methodology including quality control methods (# 41, Zheng, C 2021) as well as clarified our procedures. 

Round 2

Reviewer 1 Report

The authors responded to my concerns regarding the experimental design and the small sample size.  I still maintain this does not translate well for humans and a singular finding, largely expected, does not advance the field to any large degree.

Author Response

Thank you again for your responses and concerns. At this point the study is concluded and cannot be changed, and we do agree that it is a small pilot study which is true of many studies involving cerebrospinal fluid (CSF). However, we think this type of analysis adds valuble information to the field. For example, since ketone supplements and diets are being studied as potential Alzheimer’s treatments, our finding of a significantly higher ketone level in CSF after infusion for those with cognitive impairment is both good and important news. We have also added one more example to our manuscript which we have also reproduced below. Thanks for your consideration for this revision.  

Text added to manuscript starts after [26]: "Understanding the nuances of the glucose metabolites may help with designing diets or supplements for individuals at risk for AD, such as E4 carriers who have early brain glycose hypometabolism even prior to cognitive symptoms [26]. For example, after a glucose challenge healthy young adult E4 carriers showed a marked increase in lactate, and a pathway analysis of the plasma metabolome suggested an increase in aerobic glycolysis. Parallel work in E4-expressing astrocytes given labeled glucose also showed higher lactate synthesis and an increase in aerobic glycolysis rather than utilizing the TCA cycle [27]. To ascertain whether this was happening in the brains of human subjects, since glycolysis intermediates are largely absent from CSF as we and others have now shown, other methods would need to be deployed such as those utilizing radiolabeled glucose."